# Efficacy of Abdominal Ultrasonography for Differentiation of Gastrointestinal Diseases in Calves

**DOI:** 10.3390/ani12192489

**Published:** 2022-09-20

**Authors:** Naoaki Yoshimura, Takeshi Tsuka, Takaaki Yoshimura, Takeshige Otoi

**Affiliations:** 1Shimane Prefectural Federation Agricultural Mutual Aid Association, 748-1, Watarihashi, Izumo 693-0004, Japan; 2Laboratory of Animal Reproduction, Faculty of Bioscience and Bioindustry, Tokushima University, 2-24 Shinkura, Tokushima 770-8501, Japan; 3Clinical Veterinary Sciences, Joint Department of Veterinary Medicine, Faculty of Agriculture, Tottori University, 4-101, Koyama-Minami, Tottori 680-8553, Japan; 4Department of Health Sciences and Technology, Faculty of Health Sciences, Hokkaido University, Kitaku, Sapporo 060-0812, Japan

**Keywords:** abdominal ultrasonography, acute abdomen, calf, intestinal obstruction, to-and-fro flow

## Abstract

**Simple Summary:**

Gastrointestinal diseases represent one of the common causes of bovine acute abdomen, such as abdominal distention, abdominal pain, and cessation of defecation. In addition to the observable signs when performing routine auscultation, rectal palpation, and biochemical examinations of ruminal fluid and blood, these clinical observations can provide evidence suggestive of these diseases, but they generally result in an inconclusive diagnosis. Therefore, exploratory laparotomy is often used because it facilitates both diagnosis and therapeutic decisions. For bovines, abdominal ultrasonography is frequently utilized as a convenient imaging modality to assist accurate diagnosis and contribute to subsequent appropriate therapeutic choices for bovine gastrointestinal diseases. According to recent trends in human medicine and small animal practice, technical improvements have led to developments in the diagnostic value of abdominal ultrasonography, including scanning methods and the establishment of valuable diagnostic signs specific to a particular disease, e.g., a target sign for intussusception.

**Abstract:**

This study investigated the clinical efficacy of abdominal ultrasonography for abomasal dilation in three calves, intestinal volvulus in five calves, intussusception in one calf, and internal hernia in one calf. In the abdominal ultrasonograms of the abomasal dilation cases, this disease was commonly characterized by severely extended lumens, including heterogeneously hyperechoic ingesta without intraluminal accumulations of gas. In the animals with intestinal volvulus and intussusception, a to-and-fro flow was observed to be a common ultrasonographic characteristic that led to suspicion of an intestinal obstruction. The use of abdominal ultrasonography for five cases with intestinal volvulus gave no reason to suspect this disease, despite its efficacy in one case, based on an acutely angled narrowing. Although three of five animals with intestinal volvulus had intestinal ruptures, no ultrasonographic evidence could be obtained. When abdominal ultrasonography was used for one case with intussusception, this pathological condition could be strongly suspected, as a “target” sign was observed. This finding supported surgical intervention for this case, followed by treatment with manual reduction, resulting in a favorable outcome. In terms of the differential and definitive diagnosis for various intestinal diseases, abdominal ultrasonography may be poor at providing indicative evidence, but very helpful for confirming intestinal obstruction.

## 1. Introduction

Acute abdomen refers to pathological conditions within the abdominal cavity, inducing acute signs and potentially contributing to a life-threatening condition [1]. Various urogenital diseases, including uroliths, pyelonephritis, and uterine torsion can also give rise to bovine acute abdomen [1]. Additionally, primary or secondary ascites are a minor cause of acute abdomen [2,3]. The main etiological factors underlying bovine gastrointestinal diseases, and the main cause of acute abdomen, include enteritis, intestinal parasitism, sudden dietary changes, and the formation of intra- or extra-luminal masses, such as neoplasms and granulomas, and abscessation, although some of these are not well-known [4,5,6,7,8]. Various abomasal diseases, such as abomasal tympany, ulcer, and volvulus, when presenting in younger calves, usually occur between the ages of six and 14 weeks [3,8,9,10]. The younger calves affected with by left displacement of the abomasum were commonly several months of age, and sometimes ranged between two and three weeks of age [8,11,12]. Intestinal obstruction is one of the most common causes of bovine acute abdomen and can be differentiated into various types of intestinal involvement, such as volvulus, intussusception, internal herniation, incarceration, and strangulation [4,5,13,14,15]. The prevalence of intussusception varies between 0% and 15%, typically in calves aged less than two months [5,16,17,18,19]. The prevalence of internal herniation or incarceration has been reported to be between 0.6% and 6.4% [13,20].

Bovine acute abdomen is routinely indicated by macroscopic findings, such as abdominal pain, and various degrees of severity of abdominal distension. These signs associated with intussusception are commonly found between 24 and 48 h after the affection [1,7]. Rectal palpation enables the identification of intussusception as a hard, sausage-like structure, with constipation also apparent [7,18,21,22]. Auscultation and percussion can generate a ‘ping’ sound and that of fluid splashing in the large areas between the eighth rib and the paralumbar fossa, respectively [1]. However, the appearance of clinical signs cannot always provide sufficient diagnostic evidence to differentiate between the variety of intestinal diseases that can cause bovine acute abdomen.

Radiography is traditionally used to diagnose bovine abdominal diseases, such as peritoneal effusion and the presence of foreign bodies [23]. Bovine intestinal obstruction may be identified based on indicative findings, such as excess intraluminal gas in dilated intestinal loops or the appearance of horizontal gas–fluid interfaces [6,24,25,26,27,28]. Contrast radiography is routinely utilized to diagnose various intestinal diseases in small animals [29] and could be helpful for diagnosing abomasal dislocation and atresia coli in some bovine cases [30,31]. However, the diagnostic efficacy of radiography in bovine practice may be inferior to that in small animal practice, because of the lack of clarity of lesions due to the overlapping of the very large abdominal viscera and the presence of structural traps such as the omasum preventing the smooth intraluminal flow of contrast medium when administrated orally [27,28]. In accordance with the use of computed tomography (CT) in human medicine [32,33,34], this imaging modality has previously been used in bovine practice, as it is superior to radiography for the detection of gastrointestinal diseases but is not useful as a routine diagnostic tool [2,6,35].

Ultrasonography is the most routinely applicable imaging modality for diagnosing causative abdominal diseases in bovine cases presenting with acute abdomen. Abdominal ultrasonography enables the identification of almost all viscera, the involvement of which frequently induce bovine acute abdomen, if present at a depth of less than 10 cm from the surface of the skin during transabdominal scanning [7]. Abdominal ultrasonography is very helpful for the observation of gastrointestinal obstructions, based on abnormalities in the size, location, and intraluminal contents in the affected intestinal loops [10,19,23,27]. However, in terms of utilizing abdominal ultrasonography to definitively differentiate between the primary intestinal diseases causing intestinal obstruction, the specific ultrasonographic findings are not well understood.

The main aim of the present study was to evaluate the clinical applicability of abdominal ultrasonography to differentiate abomasal dilation (*n* = 3), intestinal volvulus (*n* = 5), intussusception (*n* = 1), and internal hernia (*n* = 1) in ten calves examined for gastrointestinal obstructions. Additionally, the clinical data for these four diseases were compared among the ten animals. The clinical data included the clinical signs of acute abdomen, the choice of conservation or surgical therapies, and their outcomes.

## 2. Materials and Methods

### 2.1. Animals

In this study, the clinical records for ten calves in Shimane prefecture, Japan, were obtained between 2018 and 2021. These ten animals comprised six Holstein calves, three Japanese-black calves, and one Holstein × Japanese-black F1 hybrid calf (Table 1). Nine were female, and one was male. The animals were aged between 2 and 197 days old when presenting with acute abdomen.

### 2.2. Clinical Signs and Physical Examinations

All animals had common abnormal physical condition, such as anorexia, loss of activity, and abdominal pains when induced by palpation (Table 1). Seven of the ten animals were constipated, when the states of defecation were evaluated by owner’s complaints and rectal palpation identifying intrarectal presence or absence of the feces. Seven of ten animals had dehydration of varying degrees of severity, when evaluated macroscopically and via a skin tent test. The animals were examined by auscultation applied dorsally-to-ventrally on both sides of their bodies, allowing a ping sound (induced by percussion) and a splashing sound to be detected within their abdomens.

### 2.3. Procedure of Abdominal Ultrasonography

Abdominal ultrasonography was conducted for each of the ten animals on the day they initially exhibited clinical signs. A portable ultrasound machine (CTS-800 KS, Kyoritsu Pharmaceutical Co., Ltd., Tokyo, Japan) and a 7.5-MHz linear transducer were used. As is routine in abdominal ultrasonography, prior to examination, non-shaved skin was sprayed with alcohol; an ultrasound gel was not used. For non-sedated animals in a standing position, abdominal ultrasonography is carried out by the application of a transducer to the surface of both the left and right abdominal walls, while moving from the cranial to the caudal area of the abdomen, with each area scanned dorsally to ventrally.

### 2.4. Evaluation of Abdominal Ultrasonography

In this study, the observation points on the abdominal ultrasonograms of each affected animal included three types of ultrasonographic signs, as previously reported [29,36]. Briefly, a to-and-fro flow suggests that the ingesta move repeatedly between the orad and distal parts within the dilated lumen of the longitudinal section of the affected intestine [29,36]. An acutely angled narrowing represents the transition parts between the dilated and obstructed loops of the longitudinal section of affected intestines [26,37,38]. A target sign or bull’s-eye sign represents the invaginated parts, in which the orad portion of the gut (intussusceptum) is engulfed within the distal portion of the gut (intussuscipiens), as a specific ultrasonographic finding of the cross-section of intussusception [6,19,22,25,39,40,41,42,43].

### 2.5. Therapeutic Methods

The results of these routine examinations were used to inform the choice of surgical or conservation therapies for nine of the animals, although one animal died suddenly following the examination. Conservation therapy included a daily intramuscular injection of metoclopramide hydrochloride (Terperan injection, ASKA Animal Health Co., Ltd., Tokyo, Japan) and a daily oral administration of a combined probiotic (*Bacillus mesentericus*) and betaine hydrochloride product (Biopair, Towa Pharmaceutical Co., Ltd., Osaka, Japan) [44,45,46]. Regarding the common procedure of laparotomy used in this study, the animal was positioned in left lateral recumbency under anesthesia from an intravenous injection of xylazine sulfate (0.2 mg/kg, xylazine injection 2% Fujita, Fujita Pharmaceutical Co., Ltd., Tokyo, Japan). The area of incision in the abdominal walls was shaved, disinfected, and anesthetized locally with a subcutaneous injection of lidocaine hydrochloride (10 mL; Xylocaine injection 2%; AstraZeneca K. K., Osaka, Japan). A 10- to 20-cm dorsal–ventral skin incision was made in the right abdominal walls. When each affected intestinal loop could be detected through surgical openings made following the incision through the muscular layers and the peritoneum, the macroscopic and palpated findings of the affected loop were used as intraoperative evidence to choose whether to use manual reduction and resection of the loop (anastomosis). Outcomes were evaluated as favorable or unfavorable for each animal based on post-therapeutic result.

## 3. Results

In this study, based on the appearance of clinical signs and the results of clinical examination and laparotomy, abomasal dilation, intestinal volvulus, intussusception, and internal hernia were diagnosed in three (Cases 1–3), five (Cases 4–8), one (Case 9), and one (Case 10) of ten animals, respectively.

In terms of the clinical signs in Cases 1–3, two animals (Case 2 and Case 3) had abdominal distention in the ventral levels of their right flanks, although there was no evidence in their left flank (Table 1 and Figure 1A). Cases 1–3 had the normal defecation and no signs of dehydration. Auscultation identified splashing sounds at regular intervals within the ventral levels of their right abdomens, despite no ping sound being induced by percussion in Cases 1–3 (Table 2). In all three cases, ultrasonography of the ventral areas of their right abdomens revealed that the ingesta were fully contained within the severely dilated abomasal lumens (Figure 2A). The abomasal ingesta were ultrasonographically characterized by a heterogeneous mixture of hypoechoic and hyperechoic contents. No intraluminal accumulations of gas were evident, as there were no ultrasonographic appearances of posterior acoustic shadowing. The areas of the pylorus could not have been demonstrated. No abnormal distention was evident in the small intestinal loops when scanned at the caudal and ventral areas of the right abdomen. Thus, these three cases (Cases 1–3) were suspected to have abomasal dilation. Conservation therapies for these animals enabled rapid improvements in their clinical signs, resulting in favorable outcomes.

In Cases 4–8, two cases (Case 4 and Case 5) had no evidence of abdominal distention, despite presenting with severe depression, resulting in them laying down (Figure 1B, Table 1). Abdominal distention of the right flank and both flanks were found in two cases (Case 6 and Case 7) and one case (Case 8), respectively (Figure 1C). Auscultation identified a splashing sound in all five animals with intestinal volvulus. A ping sound could be heard in one of five animals (Case 8) when examined by auscultation. Abdominal ultrasonography for Cases 4–8 showed for all animals findings of severe dilation of the small intestinal loops in the ventral area of their right abdomen. On the ultrasonograms showing cross-sections of the dilated loops, >2 loops could be imaged on the same screens (Figure 2B). In Case 4, on the longitudinal ultrasonogram showing the acutely curved dilated intestinal loop, the lumen was narrowed with an acute angle from the dilated, orad parts, due to gentle manual rotation of the transducer (Figure 2B and Table 2). In the four of five animals with intestinal volvulus (Cases 5–8), this finding was not observed in the ultrasonograms showing the longitudinal sections of the dilated intestinal loops. In Case 5, on a cross section of the round, dilated intestinal loop, the intestinal wall was severely thickened to approximately 5 mm throughout its whole circumference (Figure 2C). The hyperechoic contents were included within a 5-to-10-mm dilated lumen, surrounded by the thickened, hypoechoic wall. At the cross section of the maximum dilated part, the diameter was 3.5 cm (Table 2). In common with the ultrasonographic findings in Cases 6–8, multiple cross sections of the dilated intestinal loops could be observed on one screen of the ultrasonogram (Figure 2D,E). The intraluminal contents were demonstrated as hypoechoic with or without hyperechoic spots. On the ultrasonograms in Cases 6–8, no peritoneal effusion could be identified between the intestinal loops’ spaces. A to-and-fro flow was commonly demonstrated in the dilated lumens of the longitudinal sections of the intestinal loops of all five animals (Cases 4–8) (Appendix A). Additionally, ultrasonography revealed no evidence of propulsive contraction in all five animals (Cases 4–8). Based on their clinical data, for Cases 4–7, laparotomy was the chosen therapeutic option.

When performing laparotomy for Case 4, the affected region of the small intestine could be palpated as a ball-like mass, soon after inserting the surgeon’s hand through the surgical opening. Subsequently, it could be easily taken out from the peripheral normal small intestines, as there was no adhesion of the intestinal mass with the peripheral viscera (Figure 3A). In the affected intestinal loops, no discoloration was macroscopically evident. Additionally, palpation revealed the loops’ hardness and elasticity to be similar to those of the normal small intestine. In the intestinal mass, the tangled parts were carefully released, resulting in a smooth, manual reduction of loops. Macroscopically, manual reduction confirmed that the tubular structures were maintained normally within the dilated-to-narrowing transitional part when exposed from the released intestinal mass. Several minutes after manual reduction, restarted peristaltic movements and normal blood flow were identified.

When performing laparotomy for Case 5, via the surgical opening, a ball-like dilation of the intestinal loop was protruded in the center of the tangled mass of dilated intestinal loops (Figure 3B). Within this region, the intestinal loops were normal in color and elasticity and palpating allowed the detection of their slight adhesions. The affected loops were treated by manual reduction, as macroscopically they were observed to have no discoloration or ischemic change, with palpable normal hardness and elasticity. Two animals (Cases 4 and 5) recovered quickly from their digestive problems, resulting in favorable outcomes following surgery.

In Cases 6 and 7, discoloration was observed through their surgical opening, extending across the surfaces of the affected loops. In both animals, perforations were present within the discolored walls of the affected intestinal loops (Figure 3C). The yellowish ingesta flowed outward via the pathological intestinal perforations and adhered to the surfaces of the peripheral intestinal loops (Figure 3D). Following discussions with their owner, anastomosis of the affected intestine was not carried out for these two animals. These two animals were dead at one postoperative day. Case 8 died suddenly on the day of the examination. With the owner’s agreement, an autopsy was conducted and revealed a highly tangled mass comprising multiple, dilated intestinal loops that adhered tightly to the white, fibrous inflammatory tissues. Less ascites accumulation was observed around and within the intestinal mass. Reduction of the adhered tissue enabled the identification of an approximately 1-cm-sized perforation in the discolored wall of the intestinal loop located in the center of the mass, which was diagnosed as an intestinal rupture secondary to intestinal volvulus.

In Case 9, abdominal distention was not evident in either flank (Figure 1D), when abdominal pain initially presented (Table 1). The farmer noted that the animal had completely ceased to pass feces. Auscultation identified splashing sounds but no ping sounds (Table 2). Abdominal ultrasonography identified a to-and-fro flow of ingesta into the dilated lumen of the longitudinal-sectional loops. The gentle rotation of a transducer from the scanning position to the longitudinal view allowed observation of the cross-section of a 4-cm-thick, circular loop of the affected part, including 1.5-cm-thick lumen, in which the intraluminal contents could be observed as a heterogenous mixture of hyper- and hypo-echoic. The hypoechoic wall of the affected part was a double-layered structure, 5–7-mm-thick (Figure 2F). This finding was recognized as a target sign, despite the intestinal walls of the invaginated loop being slightly unclear. When performing laparotomy based on the ultrasonographic suggestion of intussusception, the affected intestinal loop can be detected between the spaces of the dilated intestinal loops via the surgical opening. Within the affected part, the engulfed, orad portion exhibited normal elasticity and hardness, despite its dark red discoloration (Figure 3E). The distal portion had no discoloration, and normal elasticity in the intestinal loop located in the orad region. Manual reduction allowed the engulfed portion to be smoothly pulled out from the invaginated portion. The animal had a favorable outcome following surgery.

In Case 10, the abdominal distention was very severe in both its flanks. The animal exhibited severe pain and no defecation. Auscultation identified ping sounds over the entire area of both sides of its swollen flanks, although no splashing sounds were heard (Table 2). Ultrasonograms of its right abdomen showed intraluminal gas within the dilated intestinal loops, measured to have a maximum diameter of 3.7 cm. Being located proximally to the skin’s surface, this generated reverberation artifacts, resulting in unclear images of the intestinal loops located more distally. Gently moving the transducer allowed observation of the dilated loops, including swirling flows of intraluminal ingesta in the spaces between the dilated, gas-filled loops. A to-and-fro flow motion was not ultrasonographically evident. Exploratory laparotomy was carried out on the day of examination. The affected intestinal loop was observed as a ball-like mass, located deeper than the multiple gas-filled dilated loops seen via the surgical opening. In the macroscopic view of the mass, the dilated loop curved acutely, making a 180° turn, followed by incarceration of the turned loop through the hole in the mesentery (Figure 3F). The reduction of the incarcerated loop was achieved following the cutting of the mesentery to make the hole wider. The incarcerated part was dark red in color. An end-to-end anastomosis was performed following resection of the discolored loop. However, the animal was dead four days after the surgery, although the reason for its death following discharge was unknown.

## 4. Discussion

In the clinical signs associated with the four types of gastrointestinal diseases targeted in this study (Table 1), different patterns of abdominal distention presented among these four diseases. The types of abdominal distention in younger affected calves can be divided into left-sided, right-sided, and bilateral [31]. In younger cases, the right-sided and bilateral types are the predominant macroscopic appearance associated with various abdominal diseases, compared with a smaller proportion of the left-sided type, induced mainly by left displacement of the abomasum as well as various diseases of the rumen [11,31]. The right-sided type is frequently found in animals with various abomasal diseases, except for left abomasal displacements, as was seen with the two of the three present cases with abomasal dilations [31]. In the five calves with intestinal volvulus, three had right-sided or bilateral abdominal distensions, followed by unfavorable outcomes due to intestinal ruptures, despite the favorable results in the remaining two cases that did not exhibit abdominal distention. Bilateral abdominal distension is one of the common clinical signs that is dependent on the degree of dilation in the obstructed intestinal loops, as well as the quantities of peritoneal effusion and leaking ingesta if ruptured [2,31]. Bilateral abdominal distention may be an important sign of an emergency situation, given the sudden death in Case 8. Additionally, Case 10 exhibited a poor therapeutic effect from the combination of anastomosis and manual reduction, which are recommended for internal herniation [7,15,20] when bilateral abdominal distention associated with excess intraluminal gas is apparent. Thus, the level of emergency in animals with intestinal obstruction may be indicated by the quantity of intraluminal gas that can be evaluated ultrasonographically, as well as through percussion and auscultation. The clinical signs of intussusception related to a poor prognosis included severe dehydration and severe abdominal distention [5]. This may explain the favorable outcome in Case 9, which presented with no abdominal distention. Increased mortality tends to be dependent on the severity of dehydration in newborn calves with acute abdomen [3,17]. The severity of dehydration may be dependent on inflammatory disturbance to intestinal absorption and the reduced or absent intake of water [19]. In this study, no strong correlation was found between the degree of dehydration and the outcome of five cases with intestinal volvulus and one case with intussusception.

In bovines, acute abdomen is a common clinical sign induced by the presence of various abdominal diseases, such as urogenital and gastrointestinal diseases [1,7]. Thus, for the clinical use of abdominal ultrasonography to differentiate these diseases, a scanning technique is required so that the entire area of both the left and right abdomen can be covered. Sonographers should operate the transducer by moving it in a ventral-to-dorsal direction parallel to the ribs, while placing it between the fifth to the thirteenth intercostal spaces and the flank on both sides [47,48]. Our scanning technique was almost identical to this, except that we performed dorsal-to-ventral scanning. We used the transabdominal scanning method of abdominal ultrasonography using a 7.5-MHz linear transducer for use on unshaved skin to which alcohol had been applied. In transabdominal ultrasonography for bovines, the use of a transducer with a low ultrasound frequency ranging from 3.5 to 5.0 MHz is recommended [23,27], although an ultrasound frequency of 7.5 MHz is suitable for the observation of peritoneal effusion [27]. When examining adult cattle, the ultrasound frequencies generated by sector, linear, or convex transducers used previously for transabdominal ultrasonography were 2.5 MHz [13,14], 2.5–5.0 MHz [49], 3.5 MHz [20,39,50,51,52,53], 3.5–5.0 MHz [54,55], and 5.0 MHz [56]. Transducers with lower ultrasound frequencies ranging between 3.5 and 5.0 MHz have also been used for the observation of various gastrointestinal diseases in younger and growing calves [8,9,48,57,58,59]. Lower frequencies of ultrasound can contribute to an increase in the detection rates of abdominal lesions, even if they are located more than 10 cm below than the skin’s surface. A 3.5 MHz ultrasound frequency has the potential to penetrate to a maximal depth of 17 cm, compared with the use of a 7.5 MHz ultrasound frequency that had difficulty in detecting lesions at a depth of more than 10 cm [22,39]. On the other hand, higher ultrasound frequencies have been selected previously, ranging from 5.0 to 7.5 MHz for scanning percutaneously while applied to the abdominal surfaces of younger calves, as this contributes to better imaging quality [2,10].

The use of abdominal ultrasonography for ten calves presenting acute abdomens could allow for a more accurate demonstration of abomasal dilation in three cases, compared with obstructions in the small intestine in seven animals. Abomasal dilation is sometimes associated with left and right displacement of the abomasum in calves aged up to six months, with previously reported prevalences of 21.5% and 11.0%, respectively [8,11]. Abomasal ulceration may occur prior to or be the cause of abomasal displacement, because facilitating decrease in the abomasal motility [12,30,31]. A reduction in abomasal motility can impair gas transport, resulting in the excess intraluminal accumulation of gas within the affected abomasum [8,12]. Thus, depending on the quantity of intraluminal gas present, abomasal displacements can be diagnosed based on ultrasonographic findings, such as the abomasal gas cap seen in the dorsal cavity and the disappearance of the lumen contents due to reverberation artifacts generated by the intraluminal gas, together with the dislocation of the affected abomasum [12,47,53,55,60]. Abomasal tympany also exhibits this gas-filled pattern [31]. However, the common ultrasonographic characteristics in the present three cases with abomasal dilation differed with this gas-filled pattern, but resembled the ultrasonographic findings in a non-displaced, dilated abomasum. The abomasal contents are imaged as homogeneously hypoechoic, as predominantly fluid-filled, and with less accumulation of gas [47,60]. This ultrasonographic pattern is distinguishable from the intraluminal findings of a normal abomasum, which appears as a heterogeneous mixture of hypo- and hetero-echogenic contents [47]. In terms of the etiology of non-displaced, dilated abomasum, functional or mechanical pyloric stenosis and small intestinal obstructions are frequently present as one of the primary lesions. Therefore, the entire area of both sides of the abdomen must be scanned [47]. Excess fluid accumulation within the abomasal lumens may result from the increased osmotic pressure of the affected abomasum as well as the inflow of extracellular fluids from the rumen [8]. Regarding the ultrasonographic characteristics of abomasal dilation in the present three cases, the intraluminal contents associated with abomasal atony appeared as echogenic mixtures without reverberation artifacts [58], despite abomasal atony causing recurrent tympany [9]. Abomasal impaction is one of the potential abomasal diseases that can be ultrasonographically observed through the intraluminal accumulation of homogenous hypoechoic intraluminal contents with multiple hyperechoic deposits. This can be caused by feeding animals poor-quality roughage and any indigestible food, with abomasal adhesion secondary to traumatic reticulo-peritonitis [3,48,59]. The transition of intraluminal echogenicity, characterized by hypoechoic fluid contents and reverberation artifacts, was ultrasonographically evident in the ventral and dorsal areas, respectively, of the impacted abomasum [59]. Additionally, a dislocated abomasum can sometimes be mistaken for a non-displaced, dilated abomasum, when observed to be homogenously hypoechoic within the ventral lumens [8,9]. Thus, scanning of the entire area of the displaced abomasum is necessary to allow observation of the gas-filled pattern within their middle and dorsal lumens [12,60]. In this study, no to-and-fro flow was evident within the dilated lumen in the ultrasonograms of the non-displaced, dilated abomasum. This sign may not appear in the lumens of the abomasum, which are much larger than the lumens of the small intestine. On the other hand, in the abdominal ultrasonograms of the larger lesions commonly found in abomasal diseases, it is difficult to visualize the entire structure in the same view [1]. This is the frequent cause of limitation in observation of the adhesion with the peripheral viscera, and dislocation of the pylorus [1]. Abdominal ultrasonography for bovines with abomasal ulcers can provide no evidence of the ulcerative lesions typically located near the pylorus area, despite effectively showing the local or diffused peritonitis associated with abomasal perforation (in type 4 abomasal ulcers) [3,10].

Abdominal ultrasonography can be helpful when choosing conservative treatments, which resulted in the healing of abomasal dilations in the present three cases, because it enabled displaced abomasum to be excluded [61]. Analgesic and anti-inflammatory drugs are recommended for pain control in bovine acute abdomen, although side-effects are possible [27]. The present three cases were treated with metoclopramide and probiotic products instead of analgesic or anti-inflammatory drugs, because they exhibited only mild abdominal pain. Metoclopramide can facilitate the stimulation of abomasal motility and has previously been used for its therapeutic effects in functional pyloric stenosis and to prevent postoperative ileus, although it has poor efficacy in calves [45]. Probiotics may facilitate the late-acting improvement of gastrointestinal health by helping to maintain a healthy state of rumen fermentation and rumen microbiota, due to their interaction with betaine hydrochloride when simultaneously supplemented orally [44,46].

The use of abdominal ultrasonography for five cases of intestinal volvulus and one case of intussusception was effective in detecting obstructions in the small intestine, when showing (1) multiple segments of dilated, tortuous intestinal loops in one ultrasonographic image; (2) the difference of echogenicity between the orad and distal portions of the lesions, respectively, including ingesta-filled and empty contents; and (3) a to-and-fro flow [3,19,29,36,40]. Additionally, reverberation artifacts are rarely observed in abdominal ultrasonograms of bovine intestinal obstructions [54]. Most animals with obstructions in the small intestine have an emergency situation, resulting in unfavorable outcomes, including their sudden death if they cannot be treated rapidly and appropriately [1,29]. From the perspective of enhancing the healing rate of obstructions in the small intestine, these indicative ultrasonographic signs can be useful for making early decisions about therapeutic options.

In human medicine, a to-and-fro flow is effectively used as an ultrasonographic sign specific to intestinal obstruction [36]. In such human examinations, the contents, such as digesta and gas, can be imaged as the repeated, intraluminal movements alternate in the orad and distal directions, without flowing into the same location of the dilated lumens within or at the oral portion of the affected intestines, regardless of the presence or absence of intestinal peristalsis [36]. In the veterinary field, this finding has previously been observed in the lumens of dilated intestinal loops in 18% of the canine cases involving intestinal obstructions [29]. In canine cases, the to-and-fro flow could be synchronized with respiratory movements, and not by peristaltic activity [29]. Abdominal ultrasonography in bovine cases identified this sign as associated with non-propulsive contraction of the affected intestines [50]. On the ultrasonograms of bovine intestinal obstructions, variation in the strength of peristaltic activity can be found ranging between absent, weak, and strong (normal) [42,50]. Thus, a to-and-fro flow may not always be found by ultrasonography within the obstructed intestinal loops. Additionally, care should be taken in the manual subcutaneous application of the transducer, as it generates the pressure facilitating this sign [36]. Doppler ultrasonography may provide useful evidence for distinguishing the to-and-fro flow of the ingesta from the flows of intraabdominal vasculatures [29].

In five present cases of intestinal volvulus, this disease was suspected in just one case, based on the ultrasonographic appearance of an acutely angled narrowing at the affected part of the small intestine, with the constricted intraluminal cavity associated with rotation around the mesenteric attachment [7]. In the orad and/or distal portions of the regional, segmental dilation of the intestinal volvulus, an excess of intraluminal gas can sometimes make their edges appear tapered on abdominal radiographs [26]. This tapered part may correspond with an acutely angled narrowing. In abdominal ultrasonograms of bovine intestinal volvulus, the echogenic and anechoic loops can be observed within the orad and distal portions of the lesions, respectively [19,29,36,38,40,49,55]. This echogenic-anechoic transition part is associated with an abrupt transition from a dilated to a collapsed loop, which is a common CT finding in human cases of intestinal obstruction [32,33]. The obstructive lesions, often found within the transition zone, can cause an abrupt tapering observed via CT, referred to as a beak sign [32,33,34]. An acutely angled narrowing may correspond with a beak sign. On the other hand, it is difficult to utilize a beak sign to distinguish the primary pathological causes of small intestinal obstruction, such as various degrees of postoperative adhesions, strangulation due to congenital or acquired bands, and intestinal volvulus, as there is no difference in findings among these diseases [33,34]. Based on the limitations of CT diagnosis [33,34], an acutely angled narrowing may be considered poor ultrasonographic evidence in terms of differentiating an intestinal volvulus from other primary intestinal diseases related to intestinal obstructions. Additionally, an acutely angled narrowing is very difficult to observe using abdominal ultrasonography, because favorable conditions to observe this sign are rarely present, such as the size and depth of the lesions causing this sign.

Intussusception can be classified into ileocolic, cecocolic, and colocolic types in 2.1%, 3.6%, and 10.7% of the bovine cases, respectively, as the majority of cases of this disease (86.4%) are involved in the small intestine [4,7,18,61]. On the other hand, the prevalence of this disease was previously reported to be uniformly distributed among the four types in younger bovine cases [18]. In one present case with intussusception, this disease was strongly indicated to be the small intestinal type through the use of abdominal ultrasonography, allowing observation of a target sign or bull’s-eye sign, in which hyperechoic contents were included within the invaginated loop on the cross-section of the affected loop [6,19,25,39,40,41,42,43,54], together with a to-and-fro flow in the ventral area of the right abdomen. The echogenicity of the intraluminal contents within the intussusceptum were previously found to vary between hyperechoic and hypoechoic [54]. This specific ultrasonographic sign can provide definitive evidence to differentiate between intussusception and intestinal volvulus, causing the clinical signs resembling between these two diseases [1,7]. However, the cross-sectional view of intestinal volvulus could be observed as a layered, circular structure, with the entire circumference thickened, resembling a target sign [39]. Thus, care should be taken to avoid overlooking and misdiagnosing this ultrasonographic sign specific to intussusception, making it difficult to distinguish it from intestinal volvulus [6,39,40].

Abdominal ultrasonography for a calf with internal herniation (Case 10) could not provide diagnostic evidence of this disease, and the associated obstructive condition of the affected loops. In Case 10, the intestinal obstruction was caused by the herniation of the intestinal loop through the pathological hole in the adjacent mesentery, although it was unknown whether the mesenteric hole was a defect or tear, or whether etiologically it formed congenitally or was acquired [15,20,39,56]. Anatomical structures related to internal herniation resulting in incarceration reportedly include rents in the greater omentum, spermatic cord, round ligament of the liver, and fibrotic bands, either formed congenitally or acquired [13,20,52,56,62]. The ultrasonographic characteristics of internal herniation are observed as anechoic fluid ingesta within the affected intestinal loops, resembling the common findings of intestinal obstruction [20,52]. While this disease may induce a to-and-fro flow, abdominal ultrasonography for Case 10 could provide no evidence of this sign, although it did allow identification of excess intraluminal gas within the multiple, dilated loops. The types of intraluminal content found within obstructed loops are generally solid and fluid ingesta, but rarely gas [47]. On the other hand, gaseous dilation may be a common finding with intestinal loops affected by internal herniation and incarceration [52,56,62]. Additionally, gaseous dilation may occur within acutely obstructed intestinal loops, such as in cases of trichobezoars or intestinal perforation [61,63]. In the present case, the excess accumulation of intraluminal gas may have been associated with the aggressive development of severe incarceration secondary to internal herniation. Reverberation artifacts generated by intraluminal gas can facilitate the disappearance of pathological findings of the deeper intestinal loops such as a to-and-fro flow [37,53]. Prior to examination with abdominal ultrasonography, percussion auscultation, inducing a ping sound, is required to detect the degree of intraluminal gas present [52,56,62,63].

In this study, three of five animals with intestinal volvulus had ruptures of the affected intestinal loops, which directly contributed to the unfavorable outcomes in the animals involved [7]. In animals with this disease, the life-threatening state is commonly dependent on developments of this disease that result in intra-abdominal outflow of the ingesta via the pathological perforations and the secondary formation of ascites [2,3,39]. If peritonitis exudates associated with intestinal ruptures are visible under ultrasonography, the various echogenic effusions (including multiple hyperechoic spots of the fibrinous deposits) can be demonstrated, thus allowing separation of the intestinal loops’ spaces [39]. However, uses of ultrasonography for the present three cases could have provided no indicative findings, due to small amounts of the leaking ingesta and ascites.

In terms of the evaluation of the degree of destructive, necrotic changes of the intestinal loops inducing their ruptures, abdominal ultrasonography is entirely inferior to exploratory laparotomy [20,56,64]. During laparotomy, macroscopic and palpation observations can provide the most reliable information for the determination of surgical options, including manual reduction and intestinal anastomosis, by allowing determination of the hardness, thickness, elasticity, and discoloration in the affected intestines, as well as pathological changes, such as mass formation and adhesion to the peripheral viscera [18,21,24,25,42,64]. In terms of the advanced technique of abdominal ultrasonography in bovine practice, Doppler ultrasonography may have the potential to assist diagnosis, therapeutic planning, and prognostic judgment, as it allows the determination of absent or poor blood flow within the ischemic intestinal loops and differentiation between to-and-fro flow and intra-abdominal blood flow [2,29,36].

Quantitative ultrasonographic assessments of small intestinal loops, such as a loop’s diameter and the thickness of the wall, can be useful for diagnosing intestinal obstructions in bovines. The diameter of the small intestinal loops is normally between 1.0 and 5.0 cm [49,55]. Differences in the normal diameters measured ultrasonographically were found in the duodenum (between 1.0 and 5.5 cm), the jejunum and ileum (between 2.0 and 4.5 cm), and the cecum (between 5.0 and 18.0 cm) [27,50]. However, there have been few reports regarding the measurements of normal small intestines in younger calves compared with adult cattle [57]. In younger calves, the ultrasonographic measurements of the normal diameters ranged between 1.8 and 2.4 cm, 0.9 and 1.1 cm, and 1.2 and 2.1 cm in the cranial and descending parts of the duodenum, the jejunum, and ileum, respectively [57]. Quantitative evaluation using CT in younger calves confirmed that the diameters of the duodenum and the other portions of the small intestine ranged between 1.4 and 5.3 cm and between 1.0 and 3.5 cm, respectively, identical to the ultrasonographic values from adult cattle [36]. The ranges of the thickness of the intestinal walls measured ultrasonographically were 2 and 3 mm in the duodenum and 1 and 2 mm in the jejunum, ileum and large intestines, corresponding with the ranges seen in adult cattle [49,55,57]. Regarding the diagnostic criteria for bovine intestinal obstruction, a common diameter was reported to be more than 3.5 cm, based on the normal ranges of healthy adult cattle [20,39,55]. The diameters of the affected loops were measured to be between 3.1 and 5.5 cm on the abdominal ultrasonograms visualizing bovine intussusception [40,41,49]. Additionally, the difference in the diameters of the obstructed intestinal loops may be dependent on the affected parts. The affected duodenum tended to have the larger intestinal diameter compared with the affected jejunum and ileum [39,51]. However, these abnormal levels were sometimes within the wide variation of normal values previously reported [36,49,55]. Additionally, the measurement accuracy is highly dependent on the skill of the sonographer [57]. Thus, ultrasonographic measurements of intestinal loops should be comprehensively evaluated alongside other associated findings, such as the number of dilated loops on the same screen [39], and the observation of various specific signs.

## 5. Conclusions

Abomasal dilation is one of the bovine gastrointestinal diseases diagnosable by abdominal ultrasonography, based on three of the present calves, despite it commonly being difficult to differentiate various abomasal diseases. A to-and-fro flow is one of the meaningful ultrasonographic signs that can indicate intestinal obstruction, as effectively used for six of seven animals with obstructions in the small intestines. However, this study indicates the low efficacy of abdominal ultrasonography for the differentiation of intestinal diseases (which was achieved in just two of seven animals with three types of intestinal diseases) and for the detection of intestinal ruptures (which was achieved in none of three animals with this condition).

## Figures and Tables

**Figure 1 animals-12-02489-f001:**
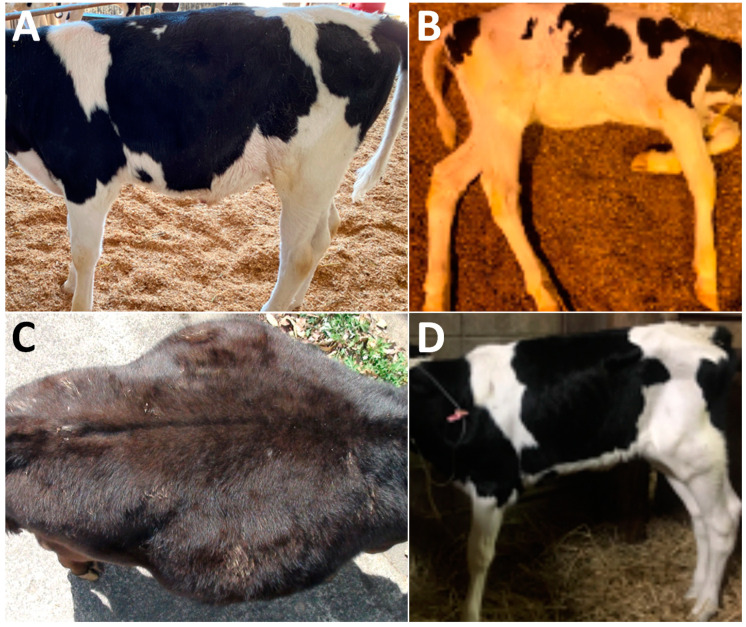
Macroscopic views of the abdomens of Case 2 (**A**), Case 4 (**B**), Case 8 (**C**), and Case 9 (**D**). (**A**) Abdominal distention is evident in the ventral region of the right flank. (**B**) Abdominal distention is not evident in either flank of the depressed, laying-down calf. (**C**) In the dorsal view, abdominal distention is markedly evident in both flanks. (**D**) Abdominal distention is not evident in either flank.

**Figure 2 animals-12-02489-f002:**
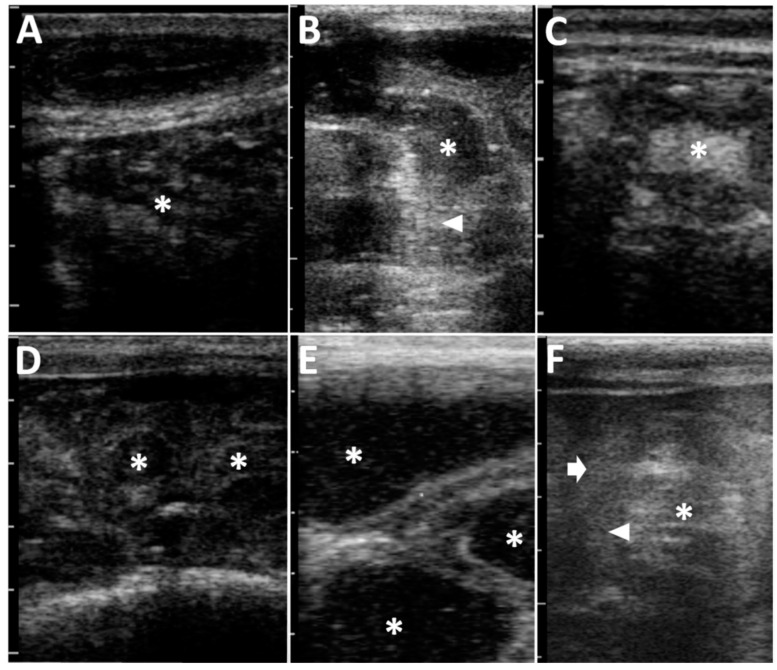
Abdominal ultrasonograms of Case 2 (**A**), Case 4 (**B**), Case 5 (**C**), Case 6 (**D**), Case 8 (**E**), and Case 9 (**F**). (**A**) The heterogeneously hypo- and hyper-echoic ingesta, including hyperechoic deposits, are seen to fully fill within the severely dilated lumen of the abomasum (asterisk). (**B**) In a longitudinal section, the dilated intestine includes the hypoechoic fluid contents with multiple hyperechoic deposits (asterisk). An acutely angled narrowing (arrowhead) is evident in the dilated, acutely curved intestinal loop. (**C**) In a cross section of the round, dilated intestinal loop, the hyperechoic contents (asterisk) are included in the thickened, hypoechoic wall. (**D**) The intraluminal contents are hypoechoic (asterisks) in multiple cross-sections of the dilated intestinal loops. (**E**) The hypoechoic contents with hyperechoic spots are included in the 1.5- to 3.0-cm-thick lumens of multiple, dilated intestinal loops (asterisk) seen on the same screen. (**F**) In the cross-section of the affected loop, the heterogenous hypo- and hyper-echoic contents (asterisk) are seen within the lumen surrounded by the thickened, hypoechoic, double, intestinal walls comprising of intussusceptum (arrowhead) and intussuscipiens (arrow). Scale bar = 10 mm.

**Figure 3 animals-12-02489-f003:**
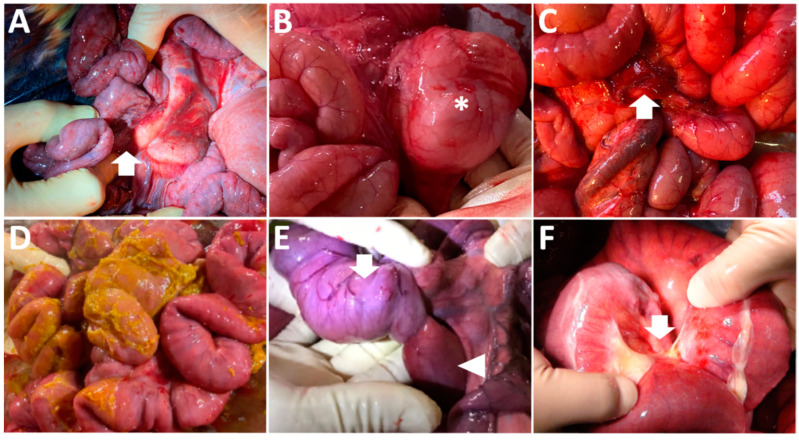
Intraoperative photos of Case 4 (**A**), Case 5 (**B**), Case 6 (**C**), Case 7 (**D**), Case 9 (**E**), and Case 10 (**F**). (**A**) The mass of the intestinal volvulus (arrow) is seen in the center of the normal intestinal loops. (**B**) A ball-like dilation of the intestinal loop (asterisk) is protruding in the center of the tangled intestinal loops. (**C**) A perforation (arrow) is evident in the center of the discolored surface of a dilated intestinal loop. (**D**) The yellowish ingesta are adhered to the surfaces of the affected intestinal loop, which has the perforation, and the peripheral loops. (**E**) The dark-red discolored portion (arrowhead) is engulfed by the invaginated, distal portion (arrow) of the affected intestinal loop. (**F**) The dilated loop seen proximal to this photo is turned, followed by running distal to such loop through the hole in the mesentery (arrow).

**Table 1 animals-12-02489-t001:** Clinical data in ten present cases.

Case	1	2	3	4	5	6	7	8	9	10
Breed ^1^	F1	Holstein	Holstein	Holstein	JB	Holstein	Holstein	JB	Holstein	JB
Sex	Female	Female	Female	Female	Male	Female	Female	Female	Female	Female
Age (day) at initial exam.	9	75	76	28	29	30	2	93	23	197
Temperature (°C)	38.4	39.0	39.3	39.3	39.0	38.7	39.0	39.4	39.5	39.0
Activity ^2^	−	−	−	−	−	−	−	−	−	−
Appetite ^2^	−	−	−	−	−	−	−	−	−	−
Abdominal pain ^2^	+	+	+	+	+	+	+	+	+	+
Abdominal distention ^2^	−	+	+	−	−	+	+	+	−	+
Defecation ^2^	+	+	+	−	−	−	−	−	−	−
Dehydration ^2^	−	−	−	+	+	+	+	+	+	+

^1^ F1 means hybrid (Holstein × Japanese-black breeds), and Japanese-black is abbreviated as JB. ^2^ Each clinical sign is evaluated as being which of detected (+) and undetected (−).

**Table 2 animals-12-02489-t002:** Examination data, diagnosis, therapy, and outcome in ten present cases.

Case	1	2	3	4	5	6	7	8	9	10
*Auscultation*										
Ping sound	−	−	−	−	−	−	−	+	−	+
Splashing sound	+	+	+	+	+	+	+	+	+	−
*Ultrasonography*										
To-and-fro flow	−	−	−	+	+	+	+	+	+	−
Target sign	−	−	−	−	−	−	−	−	+	−
Acutely angled narrowing	−	−	−	+	−	−	−	−	−	−
Propulsive contraction	+	+	+	−	−	−	−	−	−	−
Maximum loop’s diameter (cm) ^1^	ND	ND	ND	2.7	3.5	2.0	ND	4.7	4.0	3.7
Diagnosis ^2^	AD	AD	AD	IV	IV	IV	IV	IV	IS	IH
Therapy ^3^	C	C	C	L	L	L	L	N	L	L
Outcome ^4^	F	F	F	F	F	U	U	D	F	U

^1^ ND: Not detected for the measurements of maximum loop’s diameter in four cases. ^2^ Abomasal dilation, intestinal volvulus, intussusception, and internal hernia are abbreviated as AD, IV, IS, and IH, respectively. ^3^ Conservation therapy, laparotomy, and non-treatment are abbreviated as C, L, and N respectively. ^4^ E Favorable and unfavorable outcomes, and sudden death are abbreviated as F, U, and D respectively. Each clinical sign is evaluated as being which of detected (+) and undetected (−).

## Data Availability

The data presented in this study are available on request from the corresponding author.

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
