# Peer review of "Efficacy of Abdominal Ultrasonography for Differentiation of Gastrointestinal Diseases in Calves"

_animals, 2022, doi:10.3390/ani12192489_

Round 1
Reviewer 1 Report (Previous Reviewer 1)
Dear author(s)
This reviewer understands the limitations of the retrospective studies, but the present study lacks new findings.
The dose of inj, xylazine hcl used as sedative in calves appears too much higher (toxic), please check?
Author Response
Thank you for your repeatedly providing of your kindly advises.
Question: This reviewer understands the limitations of the retrospective studies, but the present study lacks new findings.
Answer: To our knowledge, “a to-and-fro flow” sign, specific to intestinal obstruction, has been described only one bovine paper “Braun and Marmier, 1995”. This study includes more detail explanation of this sign using video than one Braun’s report. Additionally, although not always, “an acute acutely angled narrowing” sign must allow indication of intestinal volvulus. To one knowledge, there was no bovine report describing this sign, despite known in human medicine.
Question; The dose of inj, xylazine hcl used as sedative in calves appears too much higher (toxic), please check?
Answer: In the revised version, the dose “2 mg/kg” is replaced by “0.2 mg/kg”.

Reviewer 2 Report (Previous Reviewer 2)
Dear Authors, my previous suggestions have been fulfilled.
Author Response
Thank you for your kindly support for creation of our manuscript.
Reviewer 3 Report (Previous Reviewer 3)
To differentiate acute gastrointestinal diseases of bovine, ultrasonography can be used simple effective modality.
This study showed the efficacy of ultrasonography in detection of intestinal obstruction of bovine.
Author Response
Thank you for your kindly support for creation of our manuscript.
Question: To differentiate acute gastrointestinal diseases of bovine, ultrasonography can be used simple effective modality. This study showed the efficacy of ultrasonography in detection of intestinal obstruction of bovine.
Answer; We agree your opinion about good efficacy of abdominal ultrasonography for diagnosing gastrointestinal diseases of bovine.
Reviewer 4 Report (New Reviewer)
The authors tried in the present study to investigate the feasibility of the abdominal as an imaging modality to diagnose and differentiate between diverse gastrointestinal conditions causing acute abdomens. However, serious points are to be considered:
1. The present investigation was conducted on calves. However, the title of the article is general. I recommend to change beef word in the title with calves.
2. The methodology and the study design are not corresponding to an original study. It tends to be more descriptive rather than providing a research-based conclusion.
3. The sample size is very limited specially that some affections were represented by one case only that is not sufficient to judge ability of ultrasonography to differentiate between different GIT affections which is the aim of the study.
Based on these pints, I recommend rejection in the present form. However, at the same time, I suggest reformulating and presenting the present study as a case series without providing a research based conclusion. This is if the journal accepts such strategy.
Author Response
Thank you for your kindly advises.
Our answers for your questions are as follows:
Question; The present investigation was conducted on calves. However, the title of the article is general. I recommend to change beef word in the title with calves.
Answer; According to advises from Reviewer 4 and academic editor, in the revised version, the title is changed as “Efficacy of Abdominal Ultrasonography for Differentiation of Gastrointestinal Diseases in Calves”
Question: The methodology and the study design are not corresponding to an original study. It tends to be more descriptive rather than providing a research-based conclusion.
Answer: The study design of this manuscript is created by reference to the previous original papers using multiple clinical cases, that already been published by the other authors for a journal “Animals”. Regarding to the description in Discussion and Conclusion, this report is designed by discussion for each of the clinical signs and ultrasonographic signs obtained from ten cases, rather than ”research-based”.
Question: The sample size is very limited specially that some affections were represented by one case only that is not sufficient to judge ability of ultrasonography to differentiate between different GIT affections which is the aim of the study.
Answer: We agree your opinion about limitation by using the results from one case (such as intussusception in Case 9, and internal hernia in Case 10). Regarding to Case 9, we could suspect strongly the affection of intussusception, based on the traditional ultrasonographic sign “target sign”. Although this sign has been reported in many previous bovine papers, we could get a meaningful evidence that a target sign was very resembled with transverse ultrasonographic section of intestinal volvulus in Case 5. These results of Cases 5 and 9 suggested that care should be taken in observation of the transverse ultrasonographic sections of the affected intestines. Thus, we believe that it is meaningful to use the data of Case 9, although one case. Regarding to Case 10, there were very few previous bovine reports about ultrasonography of internal herniaation (Lores et al., 2006; Ruf-Rits et al., 2013). These previous reports have also included no description that use of abdominal ultrasonography could facilitate complete differentiation between intestinal herniation and the other intestinal diseases. The result in Case 10 could allow indication that the artifact due to excess intestinal gas can obliterate the pathological finding of intestinal (internal) herniation on abdominal ultrasonogram, because gaseous dilation is a common finding with intestinal loops affected by internal herniation.
This manuscript is a resubmission of an earlier submission. The following is a list of the peer review reports and author responses from that submission.
Round 1
Reviewer 1 Report
Ultrasonography, no doubt is valuable non-invasive and dynamic diagnostic aid which has a very high potential for the investigation of bovine abdominal disorders.
I have following comments concerning the manuscript:
1. Sample size is too small (10 cases with 4 different conditions) that will not be sufficient to derive concrete conclusions.
2. Ultrasound scan images very poor in quality and even gross photographs are hazy. Interpretation of Fig 2B and Fig 2D is not convincing to this reviewer.
3. Ultrasonographic findings reported for diagnosing intestinal volvulus are non-specific for the intestinal obstruction due to any cause.
4. The manuscript lacks novel findings and conclusions are just repetition of already published literature.
5. Lack of control group in this study.
Reviewer 2 Report
Dear Authors,
The present manuscript presents some serious limitations:
1. The eligible criteria of the participated animals are not adequate presented. for example case 9 had 39.5 temperature. How did you rule out the possibility that this animal had pneumonia or urolithiasis.....?
2. Performing laparotomy with only xylazine is unacceptable. It is known to offer minimal analgesia

Reviewer 3 Report
Acute abdomen in bovine is important as in human. To differentiate the causes of acute abdomen, ultrasonography can be used as first line modality. This study shows it's applicability and usefulness of ultrasound in acute abdomen of bovine practice. But, showed some limitations as well. I think more cases and analyses would be needed for bovine intestinal diseases.